# The Heterostructures of CuO and SnO_x_ for NO_2_ Detection

**DOI:** 10.3390/s21134387

**Published:** 2021-06-26

**Authors:** Anna Paleczek, Bartłomiej Szafraniak, Łukasz Fuśnik, Andrzej Brudnik, Dominik Grochala, Stanisława Kluska, Maria Jurzecka-Szymacha, Erwin Maciak, Piotr Kałużyński, Artur Rydosz

**Affiliations:** 1Faculty of Computer Science, Electronics and Telecommunications, AGH University of Science and Technology, al. A. Mickiewicza 30, 30-059 Krakow, Poland; anna.paleczek@op.pl (A.P.); szafrani@agh.edu.pl (B.S.); brudnik@agh.edu.pl (A.B.); grochala@agh.edu.pl (D.G.); rydosz@agh.edu.pl (A.R.); 2Faculty of Materials Science and Ceramics, AGH University of Science and Technology, al. A. Mickiewicza 30, 30-059 Kraków, Poland; kluska@agh.edu.pl (S.K.); maria@agh.edu.pl (M.J.-S.); 3Department of Optoelectronics, Silesian University of Technology, 2 Krzywoustego Str., 44-100 Gliwice, Poland; erwin.maciak@polsl.pl (E.M.); piotr.kaluzynski@polsl.pl (P.K.)

**Keywords:** copper oxide, CuO, gas-sensing, impedance spectroscopy, NO_2_ detection, SnO_x_, tin oxide

## Abstract

Controlling environmental pollution is a burning problem for all countries more than ever. Currently, due to the increasing industrialization, the number of days when the limits of air pollutants are over the threshold levels exceeds 80–85% of the year. Therefore, cheap and effective sensors are always welcome. One idea is to combine such solutions with cars and provide real-time information about the current pollution level. However, the environmental conditions are demanding, and thus the developed sensors need to be characterized by the high 3S parameters: sensitivity, stability and selectivity. In this paper, we present the results on the heterostructure of CuO/SnO_x_ and SnO_x_/CuO as a possible approach for selective NO_2_ detection. The developed gas sensors exhibited lower operating temperature and high response in the wide range of NO_2_ and in a wide range of relative humidity changes. Material characterizations and impedance spectroscopy measurements were also conducted to analyze the chemical and electrical behavior.

## 1. Introduction

With the development of industry, technology and environmental awareness, the need to monitor emissions and levels of gases in the atmosphere increases. According to market research, the gas sensor market will develop dynamically [1,2,3]. One of the main gases with a negative impact on the environment and human health is nitrogen dioxide (NO_2_). According to the guidelines of the World Health Organization (WHO) [4], the current guideline values of 40 µg/m^3^, which corresponds to 21.3 ppb (annual average), and 100 μg/m^3^ (1 h average), which corresponds to 106.4 ppb, have negative effects on health. NO_2_, considered as an air pollutant, has several interrelated activities. Even at short-term concentrations above 200 μg/m^3^, it is a toxic gas causing inflammation in the respiratory tract [4]. NO_2_ is a source of nitrate aerosols, an important fraction of PM_2.5_, and in the presence of ozone ultraviolet light. The sources of anthropogenic NO_2_ emissions include combustion processes (heating, energy production, engines in vehicles and ships). Epidemiological studies have shown that prolonged exposure of children with asthma to NO_2_ exacerbates the symptoms of bronchitis [5]. The WHO Guidelines for Indoor Air Quality [4] provides a detailed and concise summary of studies on the effects of nitrogen oxides on human health. Moreover, a correlation was observed between the decrease in lung function and the increase in NO_2_ concentrations currently measured in the cities of Europe and North America [5]. The European Union also referred to the guideline values set by WHO in [6]. Different countries have nonidentical permissible levels of NO_2_. Various acceptable levels resulting from miscellaneous regulations and standards are not wide and therefore significant. An example of the permissible NO_2_ concentrations in the United States of America are presented in [7]. Currently, in accordance with the European Union Euro 6 standard, the permissible levels of pollutant emissions from cars, including NO_2_, have been significantly reduced; for example, NO_x_ emissions should not exceed 0.08 g/km [8,9,10]. Increasingly higher standards and requirements for exhaust emissions from means of transport require more and more accurate gas sensors necessary to control exhaust emissions. There is now more and more research on this subject. One of the many examples of car tests for exhaust emissions, especially NO_x_, is found in [11].

Many types of gas sensors are used to detect the presence of NO_2_ [12], including electrochemical sensors [13], catalytic sensors, optical sensors—infrared absorption (Infra-Red) and semiconductor sensors, which are also used in mass resonant [14,15,16,17,18]. Various materials are used in the production of semiconductor gas sensors, such as MoS_2_ [19]. Carbon based materials, such as carbon nanotubes (CNTs) [20] and graphene oxide [21], are also used for gas detection. Currently, the most popular resistance sensors are metal oxide semiconductor sensors (MOX) [13,22,23,24,25,26,27,28,29,30]. Various metal oxides are used as gas sensors; for example, barium titanate, strontium titanate and barium strontium titanate doped with various elements depending on the doped materials; they behave as p-type or n-type [31,32], n-type including zinc oxide (ZnO) [33,34], tin dioxide (SnO_2_) [35], tungsten trioxide (WO_3_) [36], indium oxide (In_2_O_3_), gallium oxide (Ga_2_O_3_), vanadium oxide (V_2_O_5_) and iron oxide (Fe_2_O_3_) and p-type metal oxides such as nickel oxide (NiO), copper oxide (CuO) [37], cobalt oxide (Co_3_O_4_), manganese oxide (Mn_3_O_4_) and chromium oxide (Cr_2_O_3_) [38,39]. As gas-sensitive materials in gas sensors, heterostructures are also used, such as tin sulfide/tin oxide (SnS_2_/SnO_2_), tungsten disulfide/titanium dioxide (WS_2_/TiO_2_), molybdenum disulfide/tin oxide (MoS_2_/SnO_2_), molybdenum disulfide/zinc oxide (MoS_2_/ZnO), reduced graphene oxide/tin oxide (rGO/SnO_2_), reduced graphene oxide/carbon dot (rGO/CD), reduced graphene oxide/molybdenum disulfide (rGO/MoS_2_) [40], reduced graphene oxide/iron oxide (rGO/F_2_O_3_) [41], cobalt tetroxide/titanium dioxide (Co_3_O_4_/TiO_2_) [42], zinc oxide/indium oxide (ZnO/In_2_O_3_), tin oxide/cupric oxide (SnO_2_/CuO), titanium dioxide/vanadium pentoxide (TiO_2_/V_2_O_5_) [30], graphene in combination with metal oxides [43] and many others.

In this paper, the gas sensors based on the heterostructures of tin oxide and copper oxide were investigated for NO_2_ detection in a wide range of concentrations. The deposited films exhibited very good selectivity to VOCs such as ethanol and acetone, and thanks to heterostructures of n-type/p-type and p-type/n-type, the influence of relative humidity was compensated. Additionally, measurements were made by impedance spectroscopy in order to determine the parameters of the equivalent electrical model for the tested nanomaterials.

## 2. Materials and Methods

### 2.1. Gas Sensor Substrates

As gas sensor substrates, a commercially available CC2 BVT alumina-based sensor substrate with electrodes were used (Figure 1a,b). After the cleaning process in an ultrasonic bath, the gas sensor substrates were placed in the deposition chamber and gas-sensitive layers were deposited as described in the sections below and schematically presented in Figure 1c.

### 2.2. Gas-Sensitive Layer Deposition

The gas-sensitive layers were obtained in the commercially available magnetron sputtering deposition system from Kurt J. Lesker Company (Hastings, East Sussex, UK), in details presented and discussed [44]. Briefly, the Ultra High Vacuum multitarget deposition system was equipped with the EpiCentre Right-angle (ECR) manipulator, which enables the deposition with the utilization of the glancing angle deposition (GLAD) technique, where deposition angle, rotation and deposition temperature are crucial parameters and can be changed by changing the ECR manipulator properties (Figure 1c). Thanks to the multitarget feature, copper and tin metal targets were used during the deposition in reaction mode without changing the deposition conditions. Such realization enables the deposition of heterostructures without any interlayer between both semiconductors.

#### 2.2.1. Tin Oxide

The tin oxide thin films were deposited in DC MF (direct current medium frequency) mode from Sn metallic target (purity 4 N—99.99%) by applying reactive sputtering under a mixture of 34% argon and 66% oxygen (purity of gases 5 N—99.999%). The deposition process parameters have been kept as in the previously reported paper, where single tin oxide-based gas-sensing layers were deposited [35,45]. In brief, the base vacuum and deposition vacuum were 1 × 10^−5^ mbar and 2 × 10^−2^ mbar, respectively. The deposition temperature was set to 200 °C and deposition time was adjusted to deposit various thicknesses with a constant power of 50 W [35,45].

#### 2.2.2. Copper Oxide

The copper oxide thin films were deposited in the same DC MF (direct current medium frequency) mode as tin oxide (Cu target purity 4 N—99.99%), and under the same pressure and gas-mixture condition (purity of gases 5 N—99.999%), which are different in comparison with those previously reported [37,46]. However, thanks to the previous experiments with Emission Optical Spectroscopy [47], the reactive atmosphere can be controlled to provide stable and repeatable conditions.

### 2.3. Gas-Sensing Measurements

Measurements of the sensor properties of the prepared samples were performed on a dedicated setup, consisting of a measuring chamber with a volume of about 300 cm^3^, with the possibility of regulating and stabilizing the temperature of the sensor in the chamber (previously presented in [48]). The target gases were supplied to the chamber via MFC controllers from gas canisters (Air Liquid, Cracow, Poland) with various initial concentrations ranging from 0.5–20 ppm. The measurement of changes in the resistance of the gas sensor was carried out using a Keithley 6517 electrometer operating with a constant test voltage of 1V. The measuring station also enables the measurement of frequency characteristics using the Solartron ModuLab XM MTS system. The measurement data were collected via a dedicated software.

#### 2.3.1. Measurements of Resistance Changes with Direct Current (DC)

Each sample underwent the same measurement procedure, which first consisted of heating and stabilizing the sample at 400 °C for 12 h. The next step, after cooling the sample to the room temperature (RT), was the measurement aimed at determining the optimal operating temperature. This measurement was performed in the temperature range from RT to 405 °C, at a relative humidity (RH) equal to 50%. The total gas flow through the chamber was 500 sccm. The concentration of NO_2_ applied was 20 ppm. The temperature was changed abruptly every 5 measuring cycles of 15 min/15 min (air/air + NO_2_). The results of this test are shown in the Figure 5a. Both samples responded best to the presence of NO_2_ at optimal temperature.

When describing their results, the authors of the research often use the terms sensitivity and response of a gas sensor interchangeably. One of the basic parameters characterizing the gas sensor is its reaction to the gas that the sensor is to detect. Most commonly used to describe a sensor’s reaction to gas is the sensor response or sensitivity. Currently, there is no uniform definition of the sensitivity (or response) of a gas sensor. Typically, the sensitivity/response (*S*) can be defined as *R*_0_/*R_g_* for reducing gases or *R_g_*/*R*_0_ for oxidizing gases, where *R*_0_ is the resistance of the gas sensors in the reference gas (usually air) and *R_g_* is the resistance in the reference gas containing the target gases. Both *R*_0_ and *R_g_* have a significant relationship with the surface reactions taking place [49]. In this paper, this definition was used to determine the sensor’s response to NO_2_:(1)$$S=\frac{R_{0}}{R_{g}}$$
where $R_{0}$ is the sensor resistance without gas presence and $R_{g}$ is the sensor resistance in the presence of gas.

Another common definition of sensitivity are the dependencies [50,51]:(2)$$S=\frac{R_{0}}{R_{g}}-1$$
(3)$$S=\frac{\left|R_{0}-R_{g}\right|}{R_{g}}$$

Scientists have different definitions of sensitivity, and the very concept of sensitivity is often used interchangeably with the sensor response. The definition adopted in this work, given by Formula (1) for the sensor’s lack of response to the presence of NO_2_ (no change in resistance), results in a value equal to 1. However, the greater the sensor’s response to gas, the higher the ratio (1) gives the result. When defining the sensor response given by Formula (1), the result is always equal to (no gas response) or greater than 1.

#### 2.3.2. Measurements of Electrical Properties with Alternating Current (AC)

Measurements of electrical properties by means of direct current methods are strongly disturbed by the polarization phenomena occurring in the measuring system [52]. These phenomena usually overestimate the value of the measured resistances, and in the case of low-voltage measurements, the current flow may be blocked, which makes it impossible to determine the electrical parameters of the sample [52,53,54]. In this case, the sample parameters calculated by the direct current methods are the sum of the electronic conductivity processes of individual grains, the polarization processes of the intracrystalline parts and the electrode processes. For this reason, in the tested material, in order to distinguish individual parameters of each of the above-mentioned processes, which are characterized by different time constants, alternating current (AC) methods are used, e.g., impedance spectroscopy [52,53,54].

This method allows the understanding of the physical and chemical processes that influence the behavior of metal oxide semiconductor gas sensors, but require correct data interpretation.

The Solatron ModuLab XM measurement system was used to determine the parameters of the studied heterostructures. This system enables the determination of impedance characteristics in a wide frequencies range. The tests carried out measurements in the range from 10^−1^ Hz to 10^6^ Hz at an operating temperature of 275 °C in RH = 50%. The (AC) signal amplitude was set to 1 V. The recorded impedance spectra were analyzed using the ZView software.

### 2.4. Material Characterization

The structural analysis of the films was carried out by an X-ray diffraction technique using PANalytical X’Pert Pro MDP with CuK*α* (λ = 1.5406 Å) at a step size of 0.04° over the 2Ɵ range of 30–80°. The chemical composition of the films was confirmed by Energy Dispersive X-ray (EDS) analysis using a FEI Inspect S50 scanning electron microscope (SEM) and an X-ray energy dispersive spectrometer with Detector EDS Octane Elect Plus and Analyzer EDAX Z2-i7. In order to assure EDS measurement, the basic SEM operation parameters were that the working distance was 10 mm, the acceleration voltages of the incident electron were 5 kV and 30 kV, the electronic beam spot size was 5 and the current intensity of the incident electronic beam was about 95 µA. Raman spectroscopy was used as a method to characterize the material tested. The Raman spectra were tested by HORIBA LabRAM HR Raman microscope with a laser treatment at 488 nm.

## 3. Results

### 3.1. Characterization Results

The crystallographic structure of the deposited gas-sensing layers was determined with X-ray diffraction (XRD) with the special adapter to the XRD system, due to the very thin thicknesses of the copper and tin oxides. The thickness was measured by the utilization of X-ray reflectance (XRR) technique and further confirmed by the utilization of a mechanical profilometer. The estimated thickness was 85 nm ± 1 nm. Figure 2 shows the XRD experiment results, where peaks from CuO are observed (110, −111, 111, 020, 022, 220) from JCPD 01-072-0629 and no peaks from SnO_x_, which suggests that SnO_x_ was not crystallized. The obtained results are in accordance with the previous results, where a higher temperature during the magnetron deposition process is required to obtain crystalline forms [55]. Moreover, the intensity of the CuO peaks is higher when CuO was deposited as a second layer, which suggests that a not-planar shape was obtained. Otherwise, the signal from the bottom layer will not be reached due to the lower X-ray measurement angle during the XRD experiments.

Regardless of the XRD measurements, to determine the composition of CuO and SnO_x_, Raman spectrometry was conducted. Raman analysis detected both crystalline and amorphous phases in the sample. Raman spectra performed at 488 nm excitation contains intense peaks attributed to glass substrate (444 cm^−1^, 783 cm^−1^, 1099 cm^−1^); two peaks at 297 cm^−1^ and 361 cm^−1^ can be assigned to CuO [56,57,58] and the peak at 128 cm^−1^ can be attributed to Sn-O, as presented in Figure 3.

The morphology of the structures was investigated by scanning electron microscopy (SEM) operated at 5 kV and 30 kV. It can be seen that the surface of the sample looks uneven due to the alumina substrate effect. The analysis of the morphology of the structures indicates that the morphology of the Al_2_O_3_ substrate is primarily visible. The CuO/SnO_x_ structures are too thin to be visualized in a classic SEM test. The EDS method turned out to be effective in the case of the investigated thin-layer CuO/SnO_x_ structures. Detailed information about the atomic structure and the elemental distribution of SnOx–CuO stack was obtained by SEM and, in combination with EDS, for elemental analysis resolution. The EDS experiments results have shown solely the presence of Cu, Sn and O. As shown in the elemental map, obtained by EDS analysis (Figure 4), the sensing structure consists of Sn, Cu and O (explained in caption of Figure 4), indicating the uniform formation of SnO_x_/CuO.

### 3.2. Gas-Sensing Characteristics

#### 3.2.1. NO_2_

The first experiments were conducted to determine the optimal operating temperature of the samples, i.e., SnO_x_/CuO and CuO/SnO_x_. As can be observed (Figure 5a), the maximal response was obtained around 250 °C for both cases; for SnO_x_/CuO, it was 240 °C and for CuO/SnO_x_, it was 275 °C. Therefore, further experiments were conducted at the optimal operating temperatures. The calibration curves (0.5–20 ppm of NO_2_) were presented in Figure 5b, and measurement data were fitted with A_1_ − A_2_e^−kx^ formula with R^2^ = 0.97278 and 0.96837 for SnO_x_/CuO and CuO/SnO_x_, respectively. The effect of relative humidity (30%, 50%, 70%) with NO_2_ (0.5–20 ppm) is presented in Figure 5c,d. However, for both cases, increased RH level resulted in an increased response.

Within the study, the multi-test was performed to verify the stability of the samples over time. Figure 6a for the SnO_x_/CuO sample and Figure 5b for the CuO/SnO_x_ sample show the results of measurements of changes in sensor resistance for three measurement cycles 60 min/60 min (air/air + NO_2_) at different concentrations of NO_2_ in the range of 0.5–5.0 ppm. The measurements were made at the optimum operating temperatures (Figure 5a) and 50% RH. For the NO_2_ concentration equal to 5 ppm, a stability test was carried out for 10 measurement cycles of 30 min/30 min (air/air + NO_2_) at the optimal temperature and RH 50%. The results for individual samples are presented in Figure 5c,d. Figure 6c,d show the results of the stable operation tests for both samples consisting of 10 cycles of 30 min/30 min (air/air + NO_2_) each. Measurements on each of the samples were carried out for several weeks at different temperatures, RH levels and NO_2_ concentrations; during this time, no change in sensor properties of the samples was observed. Both samples exhibited good stability over time.

The impedance spectra in Nyquist representation (Z” plotted against Z’) for the prepared sample of SnO_x_/CuO and CuO/SnO_x_ in the optimal operating temperature in air and under NO_2_ exposure (0.5–5 ppm) are shown in Figure 7. It was also observed that the diameter of the circle decreased when the NO_2_ admission was increased.

Nyquist plots show the imaginary part of Z′ as a function of the real part Z′′ (Figure 7a,b). In order to develop a suitable surrogate model of an electrical circuit, samples need to be analyzed over a wide frequency range. Distribution of the absolute fit error is included in Figure 8 below the Bode and Nyquist plots. The decreasing impedance with increasing frequency is evidence of the increasing conductivity of the sample. The Appendix A contain plots of frequency characteristics of the phase, Bode representations (Appendix A), which are complementary to Figure 7a,c.

A simple replacement model was used to describe the frequency characteristics of the obtained thin films as a function of NO_2_ concentration.

Based on the results of matching the impedance spectra, a substitute model consisting of R and constant phase element (CPE) elements connected in parallel were proposed. The CPE is defined according to the following equation [53]:(4)$$Z_{\mathrm{CPE}}={\left[A{\left(j\omega \right)}^{\alpha }\right]}^{-1}$$

The constant marked with the letter *A* defines the impedance modulus. The *α*, which is in the exponent, represents the impedance angle range (from 0° to 90°). In the case where the *α* value is equal to 90°, the CPE is defined as a capacitor with a capacitance equal to *A*. In the case where *α* is equal to 0, we can assume that the element is resistance. The main effect of this function in an impedance plot is to distort the semicircle.

The CPE is a distributed element that produces an impedance having a constant phase angle in the complex plane [52]. With received characteristics, it can be concluded that the electric resistance R of tested thin films reacts to the presence of nitrogen dioxide. In the analyzed cases, the P (phase) parameter of the CPE element takes values in the range of 0.90–0.95, which correspond to the capacity. The mechanisms of electronic or ionic conduction are represented by the resistive element (R). Capacitance (C) represents the sample’s polarizability in various areas of the material structure, which takes place inside the grain, at their boundaries and at the electrodes [59].

Parameters of the electrical equivalent circuit for SnO_x_/CuO, in air: R = (1.71 ± 0.02)·10^5^ Ohm, CPE-A = (4.46 ± 0.07)·10^−11^ F, CPE-P = 0.89 ± 0.01); upon 0.5 ppm NO_2_ admission: R = (1.03 ± 0.04)·10^5^ Ohm, CPE-A = (7.20 ± 0.06)·10^−1^
^1^F, CPE-P = (0.86 ± 0.01). Parameters of the electrical equivalent circuit for CuO/SnO_x_, in air: R = (0.92 ± 0.01)·10^5^ Ohm, CPE-A = (4.38 ± 0.11)·10^−11^ F, CPE-P = 0.90 ± 0.01); upon 0.5 ppm NO_2_ admission: R = (0.47 ± 0.01)·10^5^ ohm, CPE-A = (6.93 ± 0.01)·10^−11^ F, CPE-P = (0.87 ± 0.01).

The determined parameters of the substitute model allow us to determine the influence of the participation of particular phases of the structure—granular and intergranular conductivity and electrode processes—in the gas detection mechanisms.

#### 3.2.2. VOCs

The deposited gas-sensing layers were tested for cross-sensitivity under exposure to various VOCs such as acetone, ethanol and propane, as well as under various concentrations of relative humidity. The results are given in Figure 9. As can be observed in the 0.5–2.0 ppm range, the gas sensors result in no response to VOCs. Moreover, thanks to the p-n and n-p structures, the response to various RHs is more stable in comparison with typical single structure responses of MOX-based gas sensors, i.e., CuO-based and SnO-based.

#### 3.2.3. Response and Recovery Times

Regardless of the sensitivity and selectivity, the gas sensors are also characterized by response and recovery time(s). A good method for response and recovery times calculations was given by Zhang et al. [60]. Usually, the 10–90% range of the signal changes is used for determining these times. However, within this study, the developed gas sensors were tested in the measurement setup that was designed for air-pollution monitoring in the specific industrial application which is classified. Therefore, the simple comparison between obtained responses and responses reported in other studies cannot be conducted. During the determination of the response and recovery times, the sensors were tested in cycles lasting 60 min/60 min (air/air + NO_2_). For such long measurement cycles, it is possible to assume a relatively steady state of the sensor operation after nearly one hour of operation in constant conditions of the presence or absence of NO_2_ in the measuring chamber. The concentration of NO_2_ was 5 ppm. The samples operated at the optimum operating temperature and a relative humidity of 50%. Figure 10a shows the response and recovery time determination method for the SnO_x_/CuO sample and Figure 10b shows this for the CuO/SnO_x_ sample.

Interestingly, the developed gas sensors based on SnO_x_/CuO composition exhibited both higher and faster NO_2_ response, while recovery times were comparable. Therefore, the SnO_x_/CuO structure is preferred and further experiments will be conducted to define an optimal ratio between both materials.

## 4. Conclusions

The analysis of air pollutants is currently one of the burning tasks. Thanks to technical developments, fabrication of the gas sensors based on chemo resistive structures are cheap and could be an effective way to include the sensors in many applications, including the automotive industry. However, the environmental conditions are challenging in such applications, and therefore, novel realizations are a subject of research, including heterostructures of semiconductors. In this paper, we report the research results on CuO/SnO_x_ and SnO_x_/CuO structures under exposure to various concentrations of NO_2_, as well as VOCs. The gas-sensitive layers were deposited with the utilization of the glancing angle sputtering deposition technique that offers a uniform distribution of the material; however, to provide a stable substrate for demanding conditions, such as working in the cars, the alumina substrates were used. The alumina substrates are characterized by high thermal and chemical stability, and the developed sensors have also shown good stability, selectivity and sensitivity to NO_2_ in the 0.5–20 ppm range, which covers the concentrations in the air. Thanks to the heterostructure composition, the sensors work well in the 30–70% RH range, and SnO_x_/CuO showed a higher response at lower temperatures. The drawback of the developed sensors is the higher response time; however, the measurement setup was not optimized for this issue. The authors of the paper are working on creating an electronic nose in which many gas sensors will work in a microsystem. The developed sensors showed in this article will be used as a part of the mentioned microsystem and gas-dosing parts will be further optimized. The entire electronic nose system will be presented in another article, as will other sensors included in the microsystem. Nevertheless, the obtained results are very promising.

## Figures and Tables

**Figure 1 sensors-21-04387-f001:**
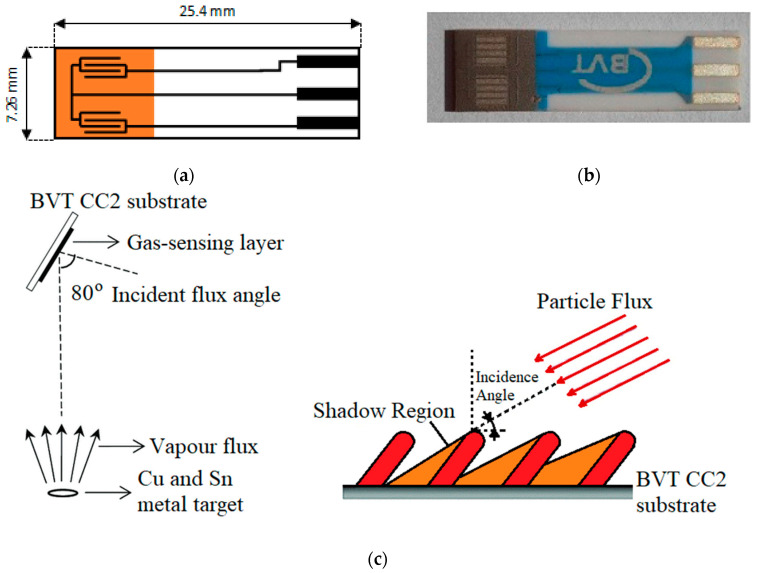
(**a**) Scheme and (**b**) picture of the CC2 BVT gas sensor substrate with deposited gas-sensing films; (**c**) the sketch of the theoretical deposition of nanostructures due to the glancing angle deposition (GLAD). Reprinted with permission from ref. [37] under CC B.Y. 4.0.

**Figure 2 sensors-21-04387-f002:**
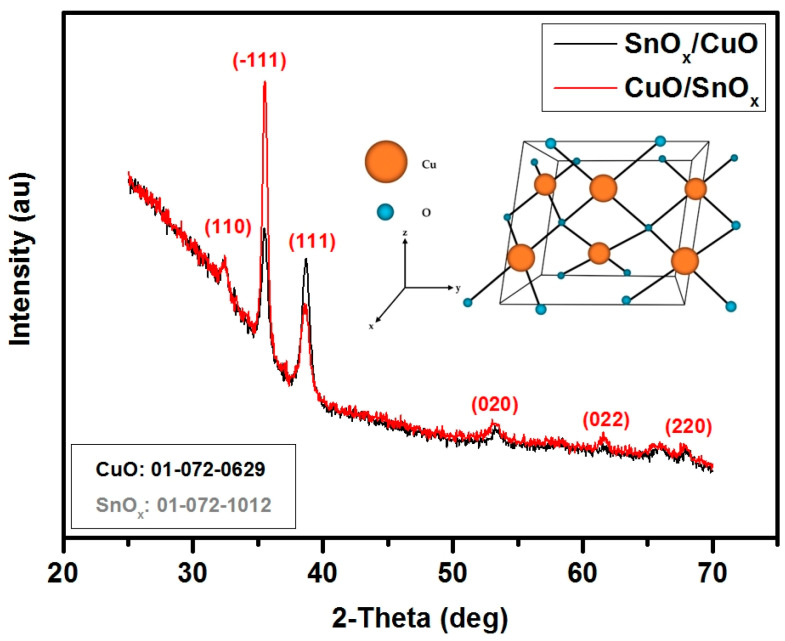
XRD patterns of deposited samples: SnO_x_/CuO and CuO/SnO_x_; inset is the CuO monoclinic structure.

**Figure 3 sensors-21-04387-f003:**
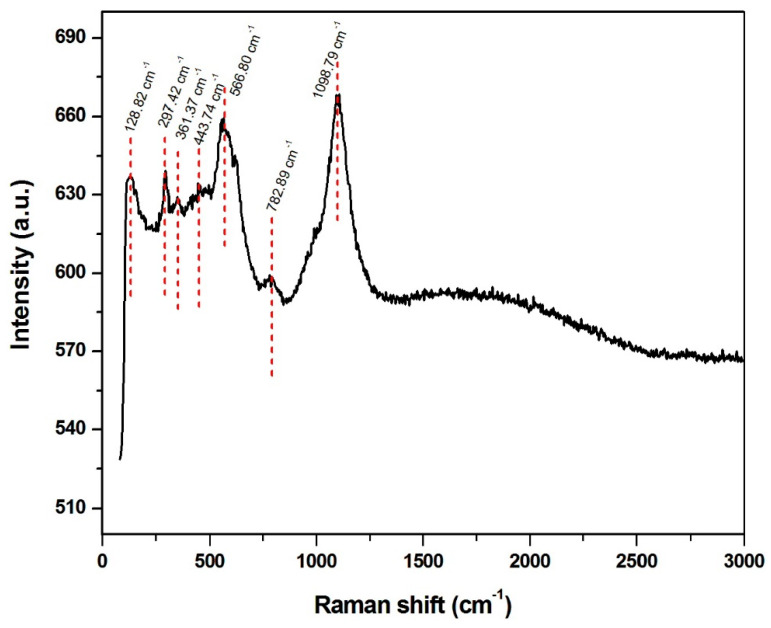
Raman spectra of the deposited sample SnO_x_/CuO.

**Figure 4 sensors-21-04387-f004:**
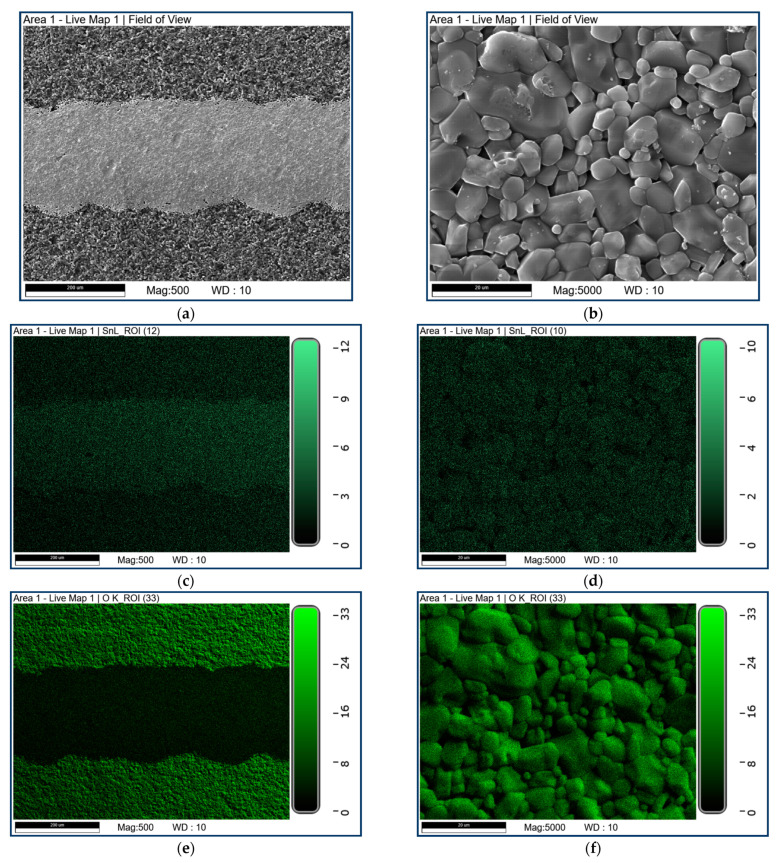
SEM and EDS results of CuO/SnO_x_ sample: (**a**) SEM image of the area from the Figure 1a where the deposited gas sensitive layer on the electrode is illustrated; (**b**) zoom of image 4a to the gas sensor material; (**c**) EDS map of Sn from image (**a**); (**d**) zoom of image 4c to the gas sensor material; (**e**) EDS map of O from image 4a; (**f**) zoom of image 4e to the gas sensor material; (**g**) EDS map of Cu from image (**a**); (**h**) zoom of image 4g to the gas sensor material. The scales indicate the counts of the EDS signal.

**Figure 5 sensors-21-04387-f005:**
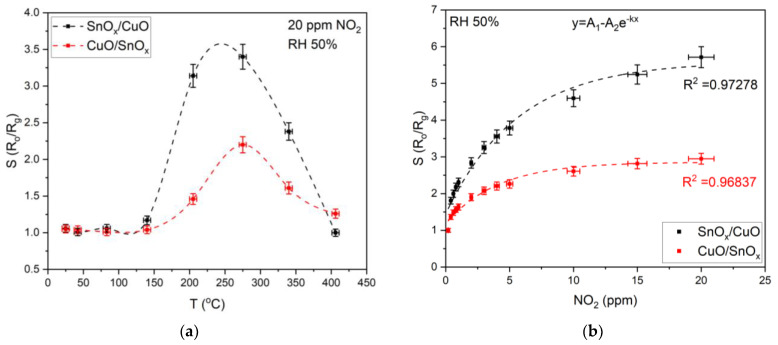
Sensor test results. (**a**) Sample response to 20 ppm NO_2_ at 50% RH as a function of temperature. (**b**) The response of samples at optimal temperature to various concentrations of NO_2_ from 0.5 ppm to 20 ppm at 50% RH. (**c**) SnO_x_/CuO sample responses to different NO_2_ concentrations depending on RH. (**d**) CuO/SnO_x_ sample responses to different NO_2_ concentrations depending on RH.

**Figure 6 sensors-21-04387-f006:**
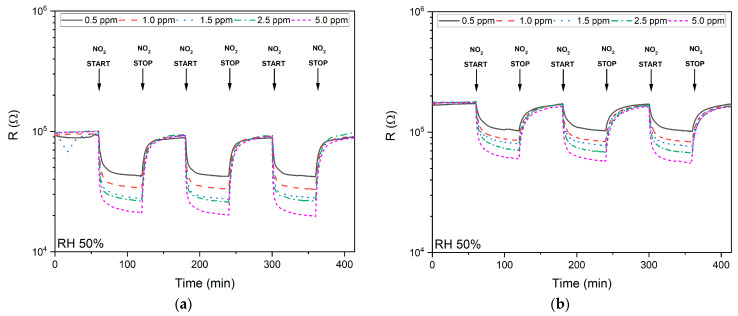
The results of measurements of the stability of the sensors. (**a**) Changes in the SnO_x_/CuO sample resistance at different concentrations of NO_2_. (**b**) Changes in the resistance of the CuO/SnO_x_ sample at different concentrations of NO_2_. (**c**) The stability of the SnO_x_/CuO sample at the optimum temperature at 10 measuring cycles. (**d**) The stability of the CuO/SnO_x_ sample at the optimum temperature at 10 measuring cycles.

**Figure 7 sensors-21-04387-f007:**
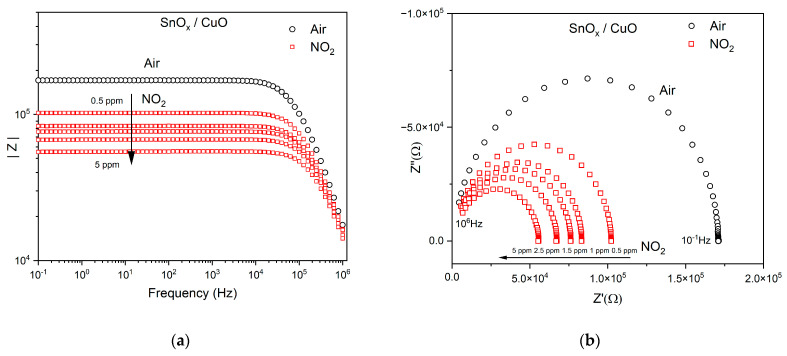
Impedance spectra of the nanomaterials in operating temperature of 275 °C in air and upon NO_2_ admission (0.5 ppm–5 ppm): (**a**) Bode representation SnO_x_/CuO; (**b**) Nyquist representation SnO_x_/CuO; (**c**) Bode representation CuO/SnO_x_; (**d**) Nyquist representation CuO/SnO_x._

**Figure 8 sensors-21-04387-f008:**
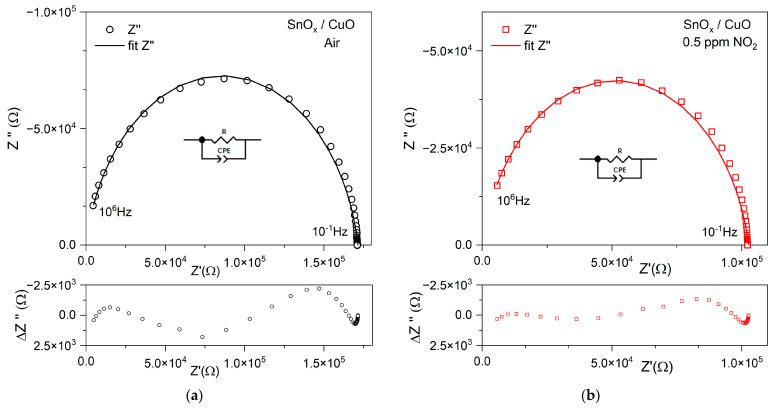
Impedance plots in Nyquist representation for SnO_x_/CuO and SnO_x_/CuO nanomaterials in measurement a very wide frequency range and electrical equivalent circuit. (**a**) SnO_x_/CuO in air atmosphere; (**b**) SnO_x_/CuO upon 0.5 ppm NO_2_ admission; (**c**) SnO_x_/CuO in air atmosphere; (**d**) SnO_x_/CuO upon 0.5 ppm NO_2_ admission. Distribution of the absolute fit error is included below the Bode and Nyquist plots.

**Figure 9 sensors-21-04387-f009:**
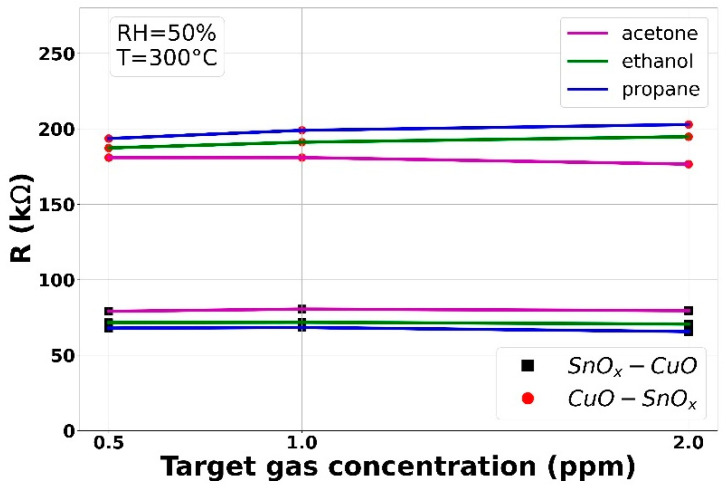
The gas-sensing results of the samples after exposure to various VOCs and relative humidity.

**Figure 10 sensors-21-04387-f010:**
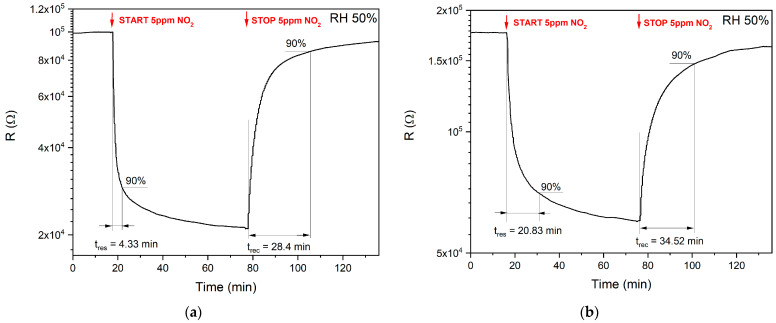
The measurement of response and recovery times. (**a**) Full 60min/60min measurement cycle (air/air + NO_2_) at RH 50% at the optimum temperature of 275 °C with response and recovery times for the SnO_x_/CuO sample marked. (**b**) Full 60min/60min measurement cycle (air/air + NO_2_) at RH 50% at the optimum temperature of 275 °C with response and recovery times for the CuO/SnO_x_ sample marked.

## Data Availability

Not applicable.

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
