# Peer review of "The Heterostructures of CuO and SnOx for NO2 Detection"

_sensors, 2021, doi:10.3390/s21134387_

Round 1

Reviewer 1 Report

Thank you to the authors. Please apply below comments to improve the paper:

-Line 21: NO2 has a different font size.

-Line 24: Keywords should be in alphabetical order.

-Please add references to support he content for lines 36-41.

-Please revise lines 44-47: repetitive words and running sentence.

-Line 56: semiconductor sensors are not necessarily known as “resistance ones”. These sensors are also used in mass resonant sensors. Please revise the sentence and add mass resonant sensors as well along with the references.

-Line 54-57: the authors first discussed the sensors based on their principle of operation and suddenly jumped into materials such as CNT and graphene oxide.

-Line 58: Authors are making a strong statement here using “ the most popular …” without providing any references. I strongly believe 3 to 4 references should be added here to support.

-Line 84: Please add what type of sensor CC2 is.

-Figure 1 caption: GLAD deposition is mentioned, but it has been defined after, in line 96.

- Line 103: terms like “DC MF” need to be defined. Please apply this comment to the entire text.

-Line 141-142: needs citation for sensor response equation.

-Line 138: the provided definition is incomplete and needs improvement.

-There are other similar sentences and statements in the entire text that need to be revised.

-There are some mistakes in terms of plural and single verbs that need to be taken care of.

-Figures’ captions are incomplete in some cases like Figure 4.

-Consistency and flow of the paper needs improvement. For example, in some cases figures are mentioned in parentheses within the text, or missing “.” at the end of the figure captions.

-Line 187: Reference 40 has a different style.

-Line 340: Authors suddenly discussed that “the developed sensors will be used as a part of the microsystems”, however microsensors and their applications and advantages are not discussed at all.

-There are several authors for this paper. Level of contribution for each author needs to be clearly discussed.

Author Response

Dear Reviewer,

thank you very much for your insightful and detailed review. All comments were very valuable and helpful.
Please see the attachment.

Sincerely

Łukasz Fuśnik

Reviewer 2 Report

The comments on this manuscript are given below, starting with line number if not indicated differently:

95 ECR was not introduced as an acronym

98 ERC was not introduced as an acronym

134 Did the temperature change spontaneously or should this sentence say "The temperature was changed..."

136 Tautology, as the optimal temperature is defined as the temperature at which the samples respond best

139 "Most commonly used to describe a sensor's response to gas is the sensor response." This sentence is self-explanatory, please rephrase.

142 Why was sensitivity defined this way? If S were defined as (R0/Rg)-1 it would be more intuitive since S=0 would indicate no response.

148 Please insert references supporting the statement that DC current methods are strongly disturbed ba the polarization phenomena.

151 Please insert references supporting the statement that these phenomena usually overestimate the value.

156 Remove full-stop after (AC).

161 English in this sentence should be corrected.

165 RH was already introduced in line 133, so writing "relative humidity" is not needed. Same for AC (introduced in line 146).

169 Overt -> Over

176 "...characterization were...", please correct either for singular or plural

182 Please introduce XRD as an acronym

185 peals -> peaks

188 CuO was deposited

196 performer -> performed

197 substrat -> substrate

198 "-1" should be written as a superscript in "...peak at 128 cm-1"

209 EDS was already introduced. Writing "energy dispersive X-ray spectroscopy" is not needed.

212 No EDS spectra are given (Counts vs. Energy) in Figure 4. If EDS was used to map the surface, please specify the meaning of color scales and mention given in c), d), e), f), g), and h).

Figure 4:     - Sub-figures a) and b) are smaller than the rest.
            - Sub-figures c), d), e), f), g), and h) overlap and distance scales are not visible.
            - Sub-figures c), d), g), and h) have low contrast and are unreadable.
            - Sub-figure h) is marked with "(h)" in bold letters, while the others are not.
            - Distance scales are too small.

218 Figure 5a is written differently than, e.g. Figure 5 (b) in line 222. Please check throughout the paper for consistent referencing of figures.

222 Why was this formula chosen to fit the data?

225 "...RH concentration..", should it state "...RH and concentration"?

Figure 5:    - Sub-figure a) shows no errors, while sub-figure b) shows them. Please make the plots consistent.
            - Sub-figures a) and b)     - shifted vertically with respect to each other.
                                    - both fitted lines look similar. Please make them different by making one of them dash-dotted, for example
                                    - same as above for c) and d), check throughout the paper.

Figure 6:    - Lines look almost the same. This is especially so if printed out. Please use different line styles for the reader to be able to discern the plots.
            - The order of magnitude can be absorbed into the axis label (1E4 R(Ohm)) to make vertical axis label more readable. Also, the authors are invited to consider using logarithmic scale where appropriate.
            - Sub-figure c) shows rising resistance from time T=0. Why is that so? Shouldn't the resistance be stable before the start of the experiment? Please explain in the text.
            - Sub-figures c) and d) - the definition of S is given earlier. Consider removing the definition from the plots to make them more readable.

Figure 7:    - Sub-figures are not the same size
            - Please make Air and NO2 symbols different, especially in Sub-figure b)
            - Bode plots usually have phase angle plotted as a function of frequency. Why is this not the case here?
            - Nyquist plots usually have one or more characteristic frequencies marked on the plot. Why is this not the case here?
            - Sub-figure d) is more readable than Sub-figure b). Please rescale b) to remove the excess empty space on top and make room for a horizontal NO2 arrow.
Figure 8:    Sub-figure d) - X axis label overlaps with values on X axis. Please shift the X axis label down and to the center. Check throughout the paper.

264 This whole section is copy-pasted from [38]. Although there might not be many ways to describe equations please rephrase to avoid possible plagiarism.

267 There are two spaces between "exponent" and "determines"

268 "cases of =1" should be rephrased. What is "=1"?

269 "with =0" should be rephrased. What is "=0"?

275 This sentence should give a range, instead it gives only a number 0.90. Should it state [0,90]?

276 This sentence is copy-pasted from [44]. Although short, please rephrase to avoid possible plagiarism.

277 The terms "polarization" and "polarizability" are different (See [44]). Please refer to: https://physics.stackexchange.com/questions/225295/what-is-difference-between-polarization-and-polarizability-and-how-do-we-define

280 Mean values and their respective errors should be reported so to give only the significant digits. Please refer to https://www.ruf.rice.edu/~bioslabs/tools/data_analysis/errors_sigfigs.html

282 "ohm" should be written as capital Greek Omega.

283 see 282

285 see 282

Figure 9: Please make labels larger.

304 If the response and recovery times are given in Figure 10 why would the classified nature of part of this research bar from comparison with other studies? For example, comparing Figure 10 from this manuscript with Figure 6 from Zhang et al. 2013 (https://doi.org/10.1016/j.snb.2013.01.082) it can be calculated that their WO3 resistive sensor responds with 0.94 1/ppm (in blue light), while sensors described in this manuscript (Figure 10 a) respond with 0.89 1/ppm during the similar 60 minute cycle of NO2.

307 Earlier in the text (e.g. line 234) cycle intervals were written without spaces in between. Please make those consistent throughout the paper.

311 "Graphically presents..." This sentence should be rephrased. For example: Figure 10 a) shows the response and recovery time determination method...

331 Insert "sputtering" between "angle" and "deposition" or present the technique in other, more specific manner.

341 Although -> Nevertheless

Author Response

(The authors gave the same response as above.)

Reviewer 3 Report

Within the paper The heterostructures of CuO and SnOx for NO2 detection  authors presented detection properties of CuO/SnOx based NO2 sensors. Paper is interesting, however the current version have to revised before the publication. Below I listed my detailed remarks:

  • I did not find novelty of presented CuO and SnO based sensor. In my opinion authors have to point out what is new in proposed construction
  • lien 21 - typo error NO2 in subscript
  • line 69 -  reduced  graphene  oxide/carbon  dot.  (rGO/CD) - is the dot(.) after dot word correct?
  • part 2.2 - please add information about purity of used metal targets and gases
  • line 156 current  (AC).  methods - an extra dot after (AC)
  • lines 195, 196, 197 - typos: and, performer, substrat
  • lines 205-206: The  CuO/SnO x  structures are too thin to be visualized in a classic SEM test - here authors mentioned that deposited films were relatively thin, in XRD section there is no information about oxide thickness. Please provide that information about CuO and SnOx thickens
  • line 219: As can be observed (Figure 5a), the maximal response was obtained around 250°C for both gases, while for SnO x /CuO it was 240°C and for CuO/SnO x  it was 275°C. - what gases authors means? Did authors used word gases instead of cases?
  • What about time stability of fabricated sensor?

Author Response

(The authors gave the same response as above.)

Round 2

Reviewer 1 Report

Thank you to the authors for implementing the comments.

The paper still needs English revision and there is no comment other than that.

Reviewer 2 Report

The authors made the effort to fully implement the suggestions and comments given after reading the first version of the manuscript.  Additionally, the answers to general questions were given in detail.

Kudos to the authors for their meticulous work. There are no further suggestions.

Reviewer 3 Report

After the corrections the paper can be published in Sensors